# The Response Surface Optimization of Supercritical CO_2_ Modified with Ethanol Extraction of *p*-Anisic Acid from *Acacia mearnsii* Flowers and Mathematical Modeling of the Mass Transfer

**DOI:** 10.3390/molecules27030970

**Published:** 2022-01-31

**Authors:** Graciane Fabiela da Silva, Edgar Teixeira de Souza Júnior, Rafael Nolibos Almeida, Ana Luisa Butelli Fianco, Alexandre Timm do Espirito Santo, Aline Machado Lucas, Rubem Mário Figueiró Vargas, Eduardo Cassel

**Affiliations:** Unit Operations Laboratory (LOPE), School of Technology, Pontifical Catholic University of Rio Grande do Sul, Av Ipiranga 6681, Building 30, Block F, Room 208, Porto Alegre ZC 90619-900, RS, Brazil; gracianenh@gmail.com (G.F.d.S.); edgar.souza@edu.pucrs.br (E.T.d.S.J.); rnolibos@gmail.com (R.N.A.); lu_fianco@hotmail.com (A.L.B.F.); alexandre.santo@acad.pucrs.br (A.T.d.E.S.); aline.lucas@pucrs.br (A.M.L.); rvargas@pucrs.br (R.M.F.V.)

**Keywords:** *Acacia mearnsii*, supercritical fluid extraction, *p*-anisic acid, response surface methodology, mathematical modeling

## Abstract

A widely disseminated native species from Australia, *Acacia mearnsii*, which is mainly cultivated in Brazil and South Africa, represents a rich source of natural tannins used in the tanning process. Many flowers of the *Acacia* species are used as sources of compounds of interest for the cosmetic industry, such as phenolic compounds. In this study, supercritical fluid extraction was used to obtain non-volatile compounds from *A. mearnsii* flowers for the first time. The extract showed antimicrobial activity and the presence of *p*-anisic acid, a substance with industrial and pharmaceutical applications. The fractionation of the extract was performed using a chromatographic column and the fraction containing *p*-anisic acid presented better minimum inhibitory concentration (MIC) results than the crude extract. Thus, the extraction process was optimized to maximize the *p*-anisic acid extraction. The response surface methodology and the Box–Behnken design was used to evaluate the pressure, temperature, the cosolvent, and the influence of the particle size on the extraction process. After the optimization process, the *p*-anisic acid yield was 2.51% *w*/*w* and the extraction curve was plotted as a function of time. The simulation of the extraction process was performed using the three models available in the literature.

## 1. Introduction

Plants are a source of natural products that have several applications and are an important raw material for obtaining compounds of interest to the pharmaceutical and food industries [1,2]. Several species of *Acacia* have been studied and their secondary metabolites have anti-inflammatory [3,4,5], antifungal [6,7,8], antibacterial [9,10,11,12], antioxidant [13,14,15], anticancer [16,17], antidepressant [18], and antifeedant properties [19], among others. The extracts of different *Acacia* species, such as *A. catechu*, *A. concinna*, *A. dealbata*, *A. decurrens*, *A. farnesiana*, and *A. senegal* are used in cosmetics, according to the International Cosmetic Ingredient Dictionary and Handbook [20].

The *Acacia mearnsii* De Wild, a member of the Leguminosae family (subfamily Mimosoideae), is widely grown in Brazil and South Africa, with an estimated 540,000 hectares cultivated worldwide [21], representing the main source of vegetable tannins. Although the biological activities of other *Acacia* genus species have already been studied and the use of its bark and wood is already common, the *A. mearnsii* flowers have not yet been industrially explored.

Supercritical fluid extraction (SFE) is an important process used to obtain bioactive compounds [22] and it stands out in the food, pharmaceutical, and cosmetic industries due to its capacity to extract compounds with a high purity without thermal degradation, as well as its use of toxic solvents [23,24]. SFE is considered a clean technology when carbon dioxide, which is considered a green solvent, is used as a solvent, offering advantages over traditional methods, such as the extraction by steam distillation and hydrodistillation methods [25,26].

In SFE, operational conditions such as temperature, pressure, and particle size influence the process efficiency [22]. The design of experiments and response surface methodologies are commonly employed for the identification and optimization of variables [27,28,29,30,31,32,33,34,35]. The methodology allows evaluating the influence of several variables regarding one or more responses with a reduced number of experiments, therefore, reducing the time and cost [36]. After the extraction process has been optimized, the mathematical modeling of the extraction process dynamic is an important step for the prediction and process scale-up [37,38]. The mathematical modeling of the extraction is substantial, to evaluate the influence of the operational parameters in the technical and economic viability of an industrial process, with a reduced number of laboratory experiments [38,39,40].

In this work, supercritical carbon dioxide extraction was applied to the *Acacia mearnsii* flowers, a part of the plant with potential that is yet to be explored. The antibacterial activity of the extract was evaluated in the crude extract and the fractions obtained by column chromatography; *p*-anisic acid was identified in the fraction with the best antibacterial activity. *p*-anisic acid, also known as draconic acid or 4-methoxybenzoic acid.

(IUPAC), is an important substance that has digestive, diuretic, and expectorant properties and it is used as an aroma component in the food and cosmetic industries as a flavor, a preservative, and an antiseptic agent [41,42]. It also has importance in medical science for the treatment of Parkinson’s disease, hepatitis B and C viruses, liver diseases, the post-radiation treatment of breast cancer, and skin desquamation [43]. Finally, it is an important substance in the production of pharmaceutical intermediates and pharmaceutical products, agrochemicals, and dyes [44]. The solubility of *p*-anisic acid in water is low, but it is highly soluble in alcohols and is soluble in ethers, as well as ethyl acetate [45].

The extraction evaluation can be carried out in terms of either the overall yield or the selectivity of a target component. The selectivity aspect is important since a high purity product for the desired analyte does not require subsequent purification operations. These purification steps make the process more expensive, in addition to exposing the extract to solvents that may not be compatible with the outcome of the product [46,47,48]. Thus, the response surface methodology was used for the optimization of *p*-anisic acid selectivity in the extract obtained by supercritical carbon dioxide extraction. The effects of pressure, temperature, and particle size were evaluated using a Box–Behnken design. For the optimized conditions within the framework investigated, the mass transfer parameters of three mathematical models were estimated to support the extraction process simulation.

## 2. Results

### 2.1. Step I—Design of the Experiments and the Chemical and Biological Evaluations

#### 2.1.1. Factorial Design 

First of all, a factorial design 2^2^ was performed evaluating the pressure and modifier effects in the global yield of the extraction process. The experiments resulted in 7 extracts with a global yield between 1.25 and 2.49% *w*/*w* (extract/plant). These results are presented in Table 1 together with the experimental design matrix, the levels of each factor, and their combinations determined by the factorial design.

The lowest yield value was obtained at the lowest pressure (120 bar), using water as the cosolvent, while the largest yield was obtained at the extraction process with the highest pressure (240 bar), using ethanol as the cosolvent. The factorial design data were processed using the Minitab^®^ statistical software and the analysis of variance (ANOVA) proved that both factors, pressure, and the cosolvent were statistically significant in the global yield of the supercritical fluid extraction. The effects were evaluated using a linear regression and the contour plot can be viewed in Figure 1. 

The model fitted to the experimental data was presented as a coefficient of determination, R², which was equal to 0.9981. The adjusted coefficient of determination had a value of 0.9944, which means that only 0.56% of the variations were not explained by the model used for the contour plot, which is presented in Equation (1):(1)$$global\;yield\;\left(\%\frac{w}{w}\right)=1.81281+0.32906\;P+0.29094\;M+0.06094\;P\;M$$
where P is the pressure and M is the modifier (values for coded variables).

#### 2.1.2. Extract Purification and Chemical Analyses 

The 7 extracts were analyzed by HPLC and in all of them, a compound with a retention time of about 10.6 min was detected (Appendix A). In the first step of this study, the compound was only analyzed qualitatively. After a comparative analysis with different standards, the most common compound was identified as *p*-anisic acid. Once the compound was identified in all extracts, the following steps involved its purification by silica gel column chromatography, thin-layer chromatography (TLC), and HPLC for each fraction.

Most of the identified compounds in the raw extracts, including the *p*-anisic acid, were only identified in the ethyl acetate fraction. Therefore, this fraction was submitted once again to column chromatography with a gradient of solvents in increasing order of polarity, and 10 fractions were collected. According to the results of the HPLC analysis, *p*-anisic acid was detected in subfractions 4 and 5, obtained with a ratio of hexane-to-ethyl acetate of 40:60 and 20:80 *v*/*v* as solvents, respectively (Appendix A). In subfraction 2, a compound that presented as orange in color in the TLC analysis (Appendix A) after staining with sulfuric vanillin was not identified in the HPLC analysis under the studied conditions (Appendix A). Nevertheless, this subfraction was chosen for the antimicrobial activity tests, along with subfraction 4, the ethyl acetate fraction, and all 7 crude extracts. We did not use subfraction 5 for the antimicrobial assays due to the lower yield compared to subfraction 4, since both have a similar chromatographic profile.

#### 2.1.3. The Antimicrobial Activity of *A. mearnsii* Supercritical Extracts

##### Bioautography 

The antimicrobial activity of the extracts was evaluated using the bioautography method against *Staphylococcus aureus* (ATCC 25923) and *Escherichia coli* (ATCC 25922). All the crude extracts inhibited the growth of the Gram-positive microorganism *S. aureus*, while none of the crude extracts inhibited the growth of the Gram-negative microorganism *E. coli*. These results are in agreement with those reported for the extracts of *Acacia podalyriifolia*, which also showed an inhibition for *S. aureus* and presented no activity against *E. coli* [49].

##### Minimum Inhibitory Concentration

The minimum inhibitory concentration (MIC) was then determined only against *S. aureus*. MIC tests were performed with the 7 crude extracts, obtained according to factorial design, and with some fractions separated by column chromatography. The best result for subfraction 4 (Table 2) was attributed to the higher *p*-anisic acid concentration verified by HPLC analyses. The activity of phenolic acids against *S. aureus* has been verified in several studies [50,51,52]. According to Basri et al. [53], the minimum inhibitory concentration (MIC) of *p*-anisic acid against *S. aureus* is MIC = 15.0.

Although the MIC of the *A. mearnsii* flower extracts obtained with supercritical fluid was high, its combination with synthetic and classic antibiotics may enhance the extract of *A. mearnsii* activity. According to studies, the synergistic effect of the association of antibiotics with plant extracts against resistant bacteria leads to new options for the treatment of infectious diseases. [54]. For example, the extracts of *Solanum paludosum Moric* obtained by the supercritical fluid process, tested by Siqueira [55], have not presented antibacterial activity but have shown modulating activity that reduced the antibiotic MIC up to eight times. Olajuyigbe & Afolayan [56] tested the effect of the methanolic extract of *A. mearnsii* bark and its synergistic effect when combined with antibiotics against 8 bacteria of clinical relevance. Concerning *S. aureus* (ATCC 6538), the minimum inhibitory concentration of the extract was 0.313 mg·mL^−1^ and it has presented synergism with antibiotics such as erythromycin, metronidazole, amoxicillin, chloramphenicol, and kanamycin. Thus, despite the low effect of *A. mearnsii* flower extracts, there is the potential for exploring its combined use with traditional drugs. These results also suggest future tests against other Gram-positive bacteria such as *Staphylococcus epidermidis*, one of the main causative agents of hospital infection [57].

### 2.2. Step II—The Optimization of SFE of p-Anisic Acid and the Mathematical Modeling of Mass Transfer

Once the *p*-anisic acid was identified in the extract, the next step was to maximize the compound selectivity due to its wide applicability. Until this point, there was no data available in the literature on the supercritical extraction of flowers from *A. mearnsii*, which justified the study.

From the Step I results, a new study regarding the extraction process conditions was evaluated. The use of ethanol as the cosolvent was maintained, considering the results obtained from the factorial design. As larger extract yields were observed for higher CO_2_ pressures set in the factorial design, the pressure range was increased from 200 to 300 bar. The solvability and diffusivity of the supercritical fluid are directly related to its density, which is a function of temperature and pressure. Thus, another factor selected for the optimization process was the temperature range, from 40 to 60 °C. The final parameter evaluated was the particle size. The flowers were ground and passed through a series of six sieves (24, 32, 42, 60, 150, and 325 mesh). Based on the amount retained in each sieve, the fractions retained in the 42, 60, and 150 mesh sieves (0.423, 0.303, and 0.125 mm, respectively) were used. From these variables, a Box–Behnken design was established and the results for the global extract yield and the *p*-anisic acid selectivity are presented in Table 3, where extractions have the solvent-to-feed ratio (S/F) equal to 61.8 g_solvent_/g_plant_.

The *p*-anisic acid selectivity ranged from 0.57 to 2.48% *w*/*w* (g *p*-anisic acid/g extract) in the crude extracts and the crude extract yield ranged from 0.86 to 7.84% *w*/*w* (g extract/g plant). The global extract yield using the Box–Behnken design had a significant increase since the highest yield was 2.49% *w*/*w* in the factorial design.

To evaluate the response surface model, an analysis of variance (ANOVA) was performed with the statistical software Minitab^®^ using the results of Table 3. The statistical significance and the influence of the extraction parameters were estimated by the analysis of variance regarding the *p*-anisic acid selectivity, which are presented in Table 4.

According to the ANOVA, only the interaction between pressure (P) and the average particle size (G) and the interaction between the temperature (T) and particle size (G) were significant (*p* < 0.05), considering a significance level of 95% (α = 0.05). The analysis also indicated that the regression was statistically significant and could be applied to describe the variation in the *p*-anisic acid amounts in the extract. However, *p*-values larger than 0.05 were obtained for the quadratic and linear regressions, indicating that part of the data behaved linearly and in a partly quadratic fashion. Thus, the regression coefficients from the response surface were estimated for the full quadratic Box–Behnken model [58], resulting in Equation (2) (for variables not coded):(2)$$\mathrm{selectivity}\;\left(\%\frac{w_{p-\mathrm{anisic}\;\mathrm{acid}}}{w_{\mathrm{extract}}}\right)=-\;9.742+8{.782\;\times \;10}^{-2}\;P+5{.495\;\times \;10}^{-2}\;T+1{.660\;\times \;10}^{-2}\;G\;-1{.429\;\times \;10}^{-4}{\;P}^{2}\;-\;1{.631\;\times \;10}^{-3}\;{\;T}^{2}\;-\;3{.523\;\times \;10}^{-5}{\;G}^{2}\;-\;1{.003\;\times \;10}^{-5}\;P\;T\;-\;1{.858\;\times \;10}^{-4}\;P\;G\;-\;7.097\;\times \;10^{-4}\;T\;G$$

The model given by Equation (2) fits the experimental data with a coefficient of determination equal to 0.9125. The validity of the model was further confirmed by the non-significant value (*p* = 0.199 > 0.05), which indicated the quadratic model as a statistically significant model for the response. Through this equation, the response surfaces shown in Figure 2 were generated. The higher selectivity of *p*-anisic acid was obtained at lower temperatures, smaller mean particle sizes, and higher-pressures, as can be observe in the Figure 2. The combination of high pressure with smaller particles led to bed compaction and preferential flow paths, so a better result was observed with high pressure and larger particle size (smaller mesh).

From the response optimizer of the Minitab^®^ software, the optimal parameters to maximize the *p*-anisic acid selectivity were defined as 278.8 bar, 40 °C, and 42 mesh and, thus, the yield would be 2.76% (Figure 3). Under these operating conditions, adjusting the pressure to 279 bar, the extraction was carried out in triplicate and the yield curves versus time were determined. The average selectivity of *p*-anisic acid was 2.51%, indicating an error of 9.06% to the value estimated by the model, confirming again the good fit. This result is also very close to the value found for extraction in the conditions of 300 bar, 40 °C, and 60 mesh (*p*-anisic acid selectivity of 2.48%) due to the slight variation in the extraction conditions. However, when working with lower pressure, there is an increase in the energy efficiency involved in the process, which is another important factor regarding process optimization.

The mass transfer mathematical modeling was performed using the three selected models and the global yield versus time curve. The experimental data and the fitted models are shown in Figure 4. The modeling was performed considering the plant particle as a sphere and its average diameter as the average particle size (mesh). 

The three models presented a good fit for the experimental data. However, as shown in Figure 4, some differences between the models were noticed. The Sovová model notes that the grinding process breaks the cell walls, making the solute easily accessible, while the Reverchon model notes that there is little solute that is easily accessible. Comparing the models with the experimental data, the Sovová model fits better in the initial extraction stage, which suggests that the grinding of *A. mearnsii* flowers increases the solute availability. The linear behavior at the beginning of extraction is observed by several authors [40,59,60,61,62,63] and is associated with the saturation of the particle surface, caused by grinding and its subsequent exposure to the extract. Despite being a simplified model, the model proposed by Crank presented a better fit with the data. Considering the good fit of the models, it is possible to say that internal diffusion controls the supercritical fluid extraction process of *A. mearnsii* flowers.

The MATLAB^®^ optimization tool was used for the Crank model fitting. The order of magnitude for the Crank model internal diffusion coefficient was the same as was found by Goto et al. [64] and Hornovar et al. [65]. The same software was also used to estimate the four parameters of the Sovová model: Z, W, $x_{k}$, and $y_{r}$, which were obtained by the least-squares method and were minimized by the Nelder–Mead simplex method [66]. Once these parameters were estimated, the mass transfer coefficients for solid and liquid phases were calculated. The solid phase mass transfer coefficient presented an order of magnitude of 10^−9^(m/s) which agrees with the values obtained by Scopel et al. [67] and Nagy et al. [68]. The mass transfer coefficient for the solvent was found to equal 9.6 × 10^−10^ m/s, whose order of magnitude is the same as found by Gallo et al. [69] when studying the supercritical extraction of *pyrethrum* flowers. The Reverchon model was implemented in the simulation software EMSO [70] and the system of equations was solved by an integrator of multiple steps, optimized by a flexible polyhedron. Thus, the values of 5.8 × 10^−4^ (s^−1^) for the internal mass transfer coefficient and 5.3 × 10^−3^ for the equilibrium constant were estimated by the least-squares method. The order of magnitude found for the parameters coincides with the values determined by Silva et al. [32], Garcez et al. [71], Almeida et al. [60], Scopel et al. [67], and Campos et al. [72]. All the parameters and the coefficients of determination for each model are presented in Table 5.

## 3. Discussion

In this work, the application of supercritical fluid extraction to obtain extracts of *A. mearnsii* flowers was studied. *A. mearnsii* flowers are a widely available feedstock since the tree is extensively cultivated due to the industrial interest in the production of tannins and wood chips. The investigation of antimicrobial activity revealed that the extracts have activity against *S. aureus*. The best result was observed for the fraction containing a higher concentration of *p*-anisic acid, suggesting a relationship between the activity and this compound. Even if the minimal inhibitory concentration values are larger than for other extracts reported in the literature, there is the potential to explore its combined use with traditional drugs. Furthermore, *p*-anisic acid is a raw material used in the cosmetic, food, and pharmaceutical industries. Thus, the extraction process was optimized to maximize the extraction of the compound of interest. 

The use of factorial and Box–Behnken designs was an efficient tool to study the influences of the process parameters in the extraction yield. The maximum global yield in the factorial design was 2.49%, while in the Box–Behnken design it was 7.84%, showing an increment of more than three times. On the other hand, this study shows that a higher global yield is not directly linked to a higher *p*-anisic acid yield.

Furthermore, the three mass transfer models fitted well with the experimental data and demonstrated that diffusion is the main mechanism associated with this process. The mass transfer parameters related to the description of the extraction process obtained in this work may be useful in future work in the scale-up and optimization of processes for obtaining the supercritical carbon dioxide extract of *A. mearnsii* flowers. Thus, this work contributes to the increasing interest in the potential use of extracts of *Acacia mearnsii* flowers. 

## 4. Materials and Methods

### 4.1. Plants

The flowers of *A. mearnsii* were supplied by Tanac S/A (RS, Brazil) and were harvested in Piratini-RS (31°17′51″ S, 53°13′29″ W) during the spring (October). The Tanac S/A is a world leader in the production of tanning plant extracts; the company has approximately 23,000 hectares of planted forests and plants around 2000 trees per hectare. All the flowers were oven-dried at 40 °C for 48 h. In the first step of the extractions, the plant material was not milled. In the process optimization with Box–Behnken design, the plant was ground in a Wiley mill. The particle size analysis was performed using a set of standard Tyler Series sieves. One hundred grams of the milled plant was added to a set of 6 sieves with 24, 32, 42, 60, 150, and 325 mesh and they were shaken for 15 min using a vibrating stirrer. The fractions with particle sizes of 42, 60, and 150 mesh were selected for the experiments [73].

### 4.2. Supercritical Fluid Extraction (SFE)

The extraction process was carried out in a pilot plant described in detail in previous works [74,75]. All experiments were conducted in the 500 mL vessels loaded with 80 g of *A. mearnsii* dried flowers. The carbon dioxide mass flow was 800 g·h^−1^ with a cosolvent flow rate of 0.5 mL·min^−1^.

### 4.3. Experimental Designs

#### 4.3.1. Factorial Design 

Initially, the supercritical fluid pressure and cosolvent effects in the global extraction yield were investigated; for that, a 2² factorial design was used. The pressure was evaluated at 120, 180, and 240 bar and ethanol, water, and a mixture of ethanol and water (1:1 *v*/*v*) were used as cosolvents, aiming to promote the extraction of polar compounds. The choice of these variables was supported by reports on the optimization of supercritical extraction [71,76,77]. In this step, the carbon dioxide temperature was 40 °C for all extractions. The experiments were performed in triplicate at the factorial design’s central point. The response surface was then associated with the regression equation. The evaluation of the effects of the variables and their interactions was achieved through an analysis of variance (ANOVA) [78]. 

#### 4.3.2. Box–Behnken Design 

The second step of this study was comprised of a Box–Behnken design, which was used for the extraction process optimization to obtain the compound, *p*-anisic acid. This experimental design consisted of a fractional factorial design method in three levels [55]. Three variables were evaluated, resulting in a total of 15 experiments. The pressure was evaluated at 200, 250, and 300 bar; the temperature was evaluated at 40, 50, and 60 °C; and an average particle size of 42, 60, and 150 mesh (0.423, 0.303, and 0.125 mm, respectively) were used. All 15 experiments used 4.7% (*w*/*w*) ethanol as the cosolvent since *p*-anisic acid is highly soluble in this solvent. These process variables were chosen based on previous studies that used RSM to analyze the SFE of plant material [32,79,80,81].

A polynomial equation was fitted into the experimental data. The optimal extraction condition was achieved in terms of the highest selectivity of the interest compound (*p*-anisic acid). The matrix design was developed in Minitab^®^ software. 

### 4.4. Chemical Analysis

To investigate the chemical composition of the extracts, each was separated by column chromatography. Both the extracts and the fractions were analyzed by thin-layer chromatography (TLC) and high-performance liquid chromatography (HPLC).

#### 4.4.1. Silica Gel Column Chromatographic Separation

Forty-two grams of the silica gel 60 (Merck) was placed into a glass column with a silica height of 15 cm. The extract was eluted with 250 mL of the following organic solvents: hexane, dichloromethane, ethyl acetate, and methanol, in increasing orders of polarity. The separation occurred in a vacuum, with 0.5 g of the dry extract [82]. In the second step, the ethyl acetate fraction was eluted in a vacuum using, as a mobile phase, the gradients shown in Table 6. The ethyl acetate fraction was chosen based on the TLC and HPLC results.

#### 4.4.2. High-Performance Liquid Chromatography (HPLC) 

The extracts were analyzed in Agilent 1200 Series liquid chromatography equipped with a U.V. detector. The separation was carried out in a C18 column (4.6 mm × 250 mm × 5 μm). The mobile phase used a binary system consisting of water (A) and acetonitrile (B), both with 2% acetic acid, in gradient mode of 80–20% of B in 30 min, with a 1.0 mL·min^−1^ flow. The injected sample volume was 5 μL and the detector was set at 258 nm. The content of *p*-anisic acid in the extracts was evaluated with a calibration curve (R² = 0.9986) containing 0.6, 0.45, 0.3, 0.15, and 0.06 mg·mL^−1^ of standard *p*-anisic acid (99% purity, Sigma Aldrich). The extracts were diluted in acetonitrile at 10 mg·mL^−1^ for their analysis by HPLC.

### 4.5. Antimicrobial Activity

The indirect bioautography method [83] was used to indicate the antimicrobial activity of the supercritical fluid extracts from *A. mearnsii* flowers. The analysis was performed to evaluate the activity against the microorganisms *Staphylococcus aureus* (ATCC 25923) and *Escherichia coli* (ATCC 25922). In this analysis, the extracts were applied to a thin layer chromatography (TLC) plate and were submitted to a run with dichloromethane as the mobile phase. After the solvent evaporation, the TLC plates were plunged into the culture medium inoculated with the microorganisms and were incubated for 24 h at 37 °C. The inoculum was prepared with a suspension of the microorganisms of 1.0 × 10^4^ CFU·mL^−1^ and was incorporated in the Mueller–Hinton agar. After the growth phase, a solution of INT (p-iodonitrotetrazolium violet) was added for better visualization of the inhibition halos [84]. Amoxicillin 0.1 mg·mL^−1^ was used as qualitative positive control while the culture medium inoculated with the microorganism was used as a negative control, without the presence of the extracts.

The antimicrobial activity was determined by the minimum inhibitory concentration (MIC) using a dilution method on microplates [85]. The microorganism inoculum was prepared with the colonies in a saline solution and the Mueller–Hinton broth, resulting in a final concentration of 1.0 × 10^4^ CFU·mL^−1^. In each microplate well, 100 µL of the Mueller–Hinton broth containing inoculums, followed by 100 µL of the extract solubilized in Tween 20, as well as water (at final concentrations of 0.75, 1.5, 3.0, 6.0, 12.0, and 24 mg·mL^−1^), were introduced. Other concentrations were used for the fractions obtained after the column chromatographic separation, defined according to each fraction weight. For the ethyl acetate, the final concentrations were 1.8, 3.6, 7.3, 14.7, 29.5, and 59.2 mg·mL^−1^; for the fraction obtained with hexane:ethyl acetate (80:20), named “subfraction 2”, the final concentrations were 2.2, 4.4, 8.9, 17.9, 35.9, and 71.3 mg·mL^−1^; and the fraction obtained with hexane:ethyl acetate (60:40), named “subfraction 4”, was tested at concentrations of 0.73, 1.4, 2.9, 5.9, 11.8, and 23.6 mg·mL^−1^.

### 4.6. Mass Transfer Mathematical Modeling

The description of the mass transfer phenomena involved in the supercritical fluid extraction was performed and evaluated using three models, as described below.

#### 4.6.1. Crank (1975) Model

The Crank Model [86] was developed from Fick’s Second Law, which considers the diffusion of a single particle in the form of a sphere. This model describes the behavior of the bed, as a whole, from the simplified mass transfer process in a single particle. The model shows that the sphere is initially at a uniform concentration and that the surface concentration is maintained as a constant. The total amount of the diffusing substance entering or leaving the sphere can be written as [86]:(3)$$\frac{M_{t}}{M_{\infty }}=1-\frac{6}{\pi^{2}}\;\overset{\infty }{\underset{n=1}{\sum }}\frac{1}{n^{2}}\exp\left(-\frac{Dn^{2}\pi^{2}t}{r^{2}}\right)$$
where $M_{t}$ and $M_{\infty }$ are the mass in a determined time and an infinite time (maximum mass obtained in the extraction), respectively, D is the diffusivity of the solute inside the particle (m^2^·s^−1^), t is the extraction time (s), r is the particle radius (m) and n is the number of the series expansion.

#### 4.6.2. Sovová (1994) Model

The model proposed by Sovová [63] considers that the extraction of the solute by supercritical CO_2_ can be divided into three periods. The first period of extraction considers that only the easily accessible solute can be extracted, which has direct contact with the solvent; the second period considers that the easily accessible solute is gradually depleted from the inlet to the outlet of the bed, and the third period includes solutes that are difficult to access, contained within the particles. Therefore, the extract mass initially present in the solid phase (O) is the sum of the easily accessible mass of solute (P) and the inaccessible mass of solute contained within the solid particles (K). The solid phase, free from the solute (N), is the constant during the extraction and relates to the initial concentrations of the solute: (4)$$x\left(t=0\right)=x_{0}=\frac{O}{N}=x_{p}+x_{k}=\frac{P}{N}+\frac{K}{N}$$

The mass balance in the fluid phase and solid phase for a bed element is described by two differential equations that were analytically solved by Sovová [63], who simplified a few hypotheses. The final expression is given by Equation (5) in terms of the extract mass relative to the extract-free solid mass: (5)$$e=\{\begin{array}{c} qy_{r}\left[1-\exp\left(-Z\right)\right]\\ y_{r}\left[q-q_{m}\exp\left(z_{w}-Z\right)\right]\\ x_{0}-\frac{y_{r}}{W}\;ln\{1+\left[\exp\left(W\frac{x_{0}}{y_{r}}-1\right)exp\left[W\left(q_{m}-q\right)\right]\frac{x_{k}}{x_{0}}\right]\} \end{array}$$
where:(6)$$q=\frac{Q\;t}{N}$$
(7)$$q_{m}=\frac{\left(x_{0}-x_{k}\right)}{y_{r}\;Z}$$
(8)$$q_{n}=q_{m}+\frac{1}{W}\;ln\frac{\;x_{k}+\left(x_{0}-x_{k}\right)\exp\left(Wx_{0}/y_{r}\right)}{x_{0}}$$
(9)$$\frac{z_{w}}{Z}=\frac{y_{r}}{w\;x_{0}}ln\frac{x_{0}\;exp\left[W(q-q_{m})\right]-x_{k}}{x_{0}-x_{k}}$$
(10)$$Z=\frac{k_{f}a_{0}\rho }{\dot{q}\left(1-\varepsilon \right)\rho_{s}}$$
(11)$$W=\frac{k_{s}a_{0}}{\dot{q}\left(1-\varepsilon \right)}$$

In the above equations, $y_{r}$ is the solubility; Z and W are the adjustable parameters for fast and slow periods, respectively, and are directly proportional to the mass transfer coefficients of each phase; the term $z_{w}$ corresponds to the boundary coordinate between fast and slow extraction; and $k_{f}$ and $k_{s}$ are the mass transfer coefficients of fluid and solid phases, respectively. The unknown quantities $x_{k}$, $y_{r}$, $k_{s}$, and $k_{f}$ were estimated by the least-squares method.

#### 4.6.3. Reverchon (1996) Model

The model proposed by Reverchon [25] was developed from a mass balance for the solid and fluid phases, according to Equations (12) and (13). In the study, the extract is considered a pseudo component, which is not readily available on the surface of the particles after the milling process. As a result, the mass transfer is controlled by internal resistance. The axial dispersion is considered negligible. The density and solvent flow rate are said to be constant along the bed:(12)$$uV\frac{\partial c}{\partial h}+\varepsilon V\frac{\partial c}{\partial t}+\left(1-\varepsilon \right)V\frac{\partial q}{\partial t}=0$$
(13)$$\left(1-\varepsilon \right)V\frac{\partial q}{\partial t}=-A_{p}k_{TM}\left(q-q^{*}\right)$$
where u is the interstitial velocity of the fluid, ε is the bed porosity, and $\rho_{s}$ is the plant density. The previous differential equations satisfy the initial conditions described by c(h,0)=0 and $q\left(h,0\right)=q_{0}$ for all h, and the following boundary conditions C(0,t)=0 for all t. A linear relationship describes the equilibrium behavior between the phases during the supercritical fluid extraction process [25]:(14)$$q^{*}=K\;C$$
where K is the volumetric partition coefficient of the extract between fluid and solid phases at the equilibrium condition. Reverchon (1996) sets the internal diffusion time as:(15)$$t_{i}=\frac{\left(1-\varepsilon \right)V}{A_{p}k_{TM}}.$$
and Equation (13) can be rewritten as:(16)$$\frac{\partial q}{\partial t}=-\frac{1}{t_{i}}\;\left(q-q^{*}\right);$$
and the internal diffusion time (Equation (17)) is related to the internal diffusion coefficient ($D_{i}$):(17)$$t_{i}=\frac{\mu \;l^{2}}{D_{i}}$$
where μ is a constant related to the particle geometry (equal to 3/5 for spherical particles) and l is the characteristic dimension given by the ratio between the particle volume and the particle superficial area.

## 5. Conclusions

The Factorial 2² design indicated that the crude extract yield was higher when ethanol was used as the cosolvent, and the maximum tested pressure (240 bar) was applied in the extraction. The crude extract obtained showed antimicrobial action against *S. aureus*. The purification by silica gel column chromatography generated a fraction rich in a compound identified as *p*-anisic acid and this fraction improved the antimicrobial performance against *S. aureus*. The use of the selectivity criterion as an objective function in the response surface method demonstrated that the condition that includes 278.8 bar, 40 °C, and 42 mesh is the condition that produces the highest amount of *p*-anisic acid per unit of extract mass. This result is important, as this optimal condition is obtained with the use of eco-friendly solvents. The three mathematical models used to simulate the extraction kinetics were adequate for the supercritical CO_2_ extraction, with aqueous ethanol as the cosolvent, from *A. meanrsii* flowers. Thus, the supercritical extraction is an adequate and clean method to obtain *p*-anisic acid from *Acacia mearnsii* flowers and this work demonstrates that the potential usage of this plant material is abundant but not exploitative. As a new source of bioactive compounds, the use of this flower extract also contributes to reducing the volume of solid waste generated by the cultivation of this plant in forests.

## Figures and Tables

**Figure 1 molecules-27-00970-f001:**
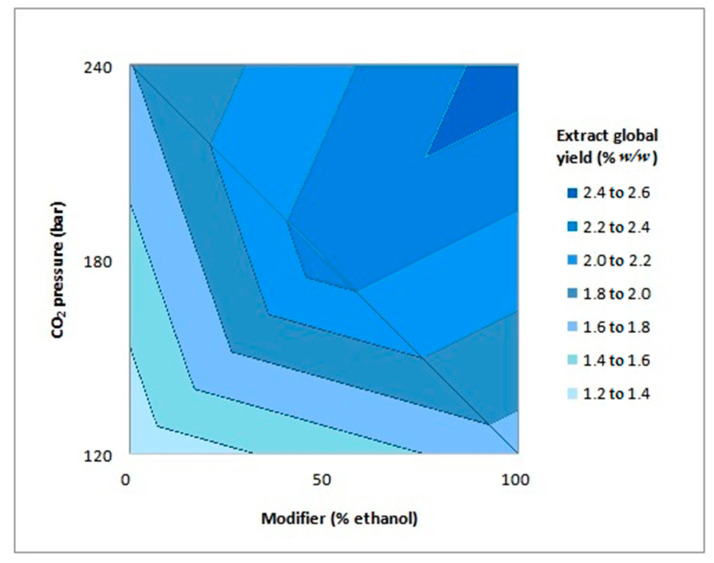
Contour plot for global extraction yield as a function of CO_2_ pressure and modifier.

**Figure 2 molecules-27-00970-f002:**
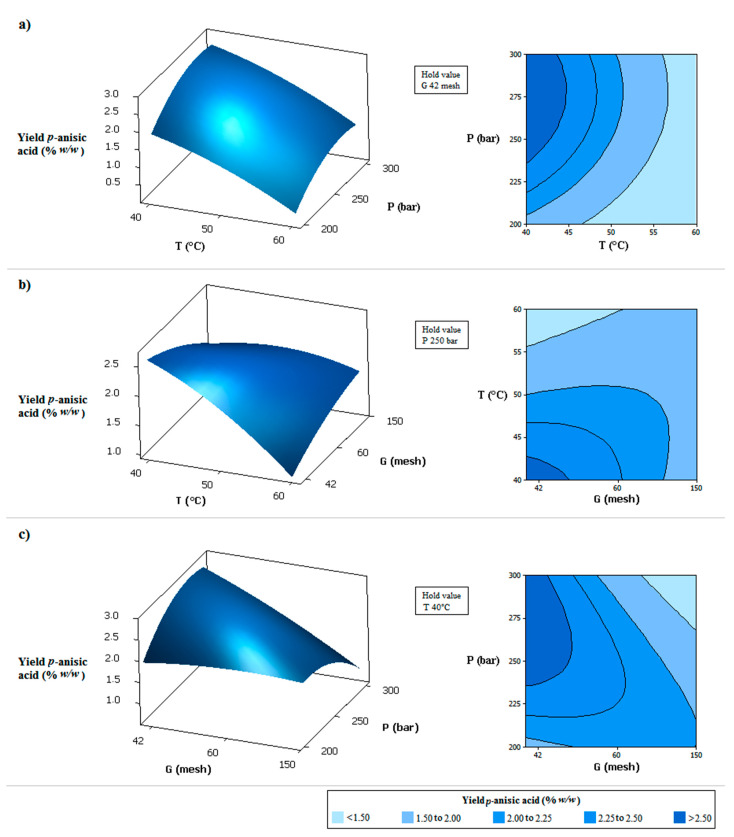
Response surfaces and contour plots for effects of two independent variables on the yield of *p*-anisic acid in the extract obtained by supercritical extraction: (**a**) CO_2_ pressure (P) and CO_2_ temperature (T); (**b**) CO_2_ temperature (T) and medium particle size of milled flowers (G); (**c**) CO_2_ pressure (P) and medium particle size of milled flowers (G).

**Figure 3 molecules-27-00970-f003:**
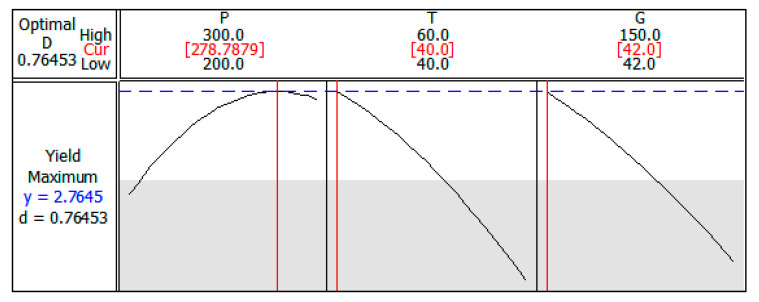
Process parameters optimized to maximum yield of *p*-anisic acid.

**Figure 4 molecules-27-00970-f004:**
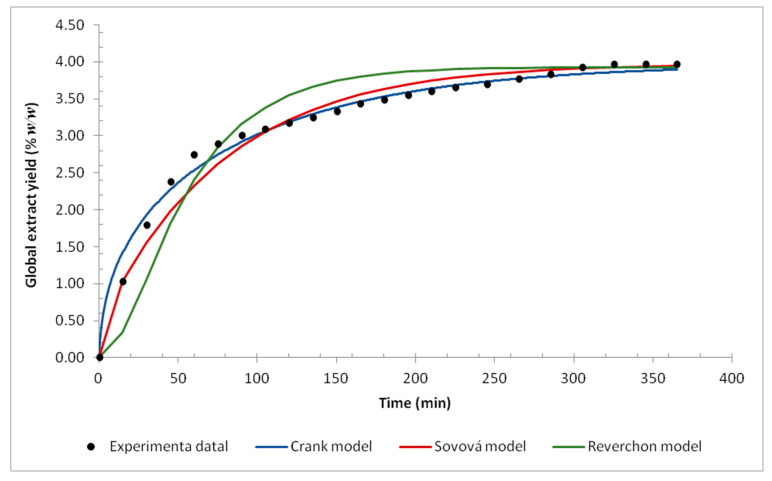
The yield curve for supercritical fluid extraction at 40 °C, 279 bar, and milled flower (42 mesh): mathematical models and experimental data.

**Table 1 molecules-27-00970-t001:** Factorial 2² design matrix and observed responses.

Standard Run Order	Codified Variables		Uncodified Variables		Global Extract Yield (% *w*/*w*) ^a^	S/F ^b^ (g_solv._/g_plant_)
	Pressure	Modifier	Pressure (bar)	Modifier		
1	−1	−1	120	Water	1.25	24.2
2	−1	1	120	Ethanol	1.70	24.0
3	1	−1	240	Water	1.71	24.2
4	1	1	240	Ethanol	2.49	24.0
5	0	0	180	Water:Ethanol (1:1 *v*/*v*)	2.27	24.1
6	0	0	180	Water:Ethanol (1:1 *v*/*v*)	2.33	24.1
7	0	0	180	Water:Ethanol (1:1 *v*/*v*)	2.30	24.1

^a^ gram of crude extract from 100 g of dried flowers; ^b^ S/F is the solvent-to-feed ratio.

**Table 2 molecules-27-00970-t002:** Minimal inhibitory concentration (MIC) of *A. mearnsii* crude extracts and its purified fractions against *S. aureus*.

Sample	MIC (mg·mL^−1^)
Extract 2 (P = 120 bar; cosolvent: ethanol)	24
Extract 7 (P = 180 bar; cosolvent: ethanol:water 1:1 *v*/*v*)	24
Extracts 1, 3, 4, 5, and 6	>24
Ethyl acetate fraction	59.2
Subfraction 2 (solvent: hexane:ethyl acetate 80:20)	35.9
Subfraction 4 (solvent: hexane:ethyl acetate 40:60)	11.8

**Table 3 molecules-27-00970-t003:** Design matrix in the Box–Behnken model and observed responses.

Run Order	Uncodified Variables			Responses	
	Pressure (bar)	Temperature (°C)	Medium Particle Size (mesh)	*p*-Anisic Acid Yield (% *w*/*w*) ^a^	Global Extract Yield (% *w*/*w*) ^b^
1	250	50	60	1.83	1.76
2	200	60	60	0.57	7.64
3	300	40	60	2.48	2.98
4	200	40	60	2.19	5.11
5	250	50	60	2.11	6.65
6	250	60	150	1.86	4.45
7	300	50	150	0.82	2.51
8	200	50	150	1.96	2.49
9	200	50	42	1.21	4.85
10	300	60	60	0.84	3.26
11	300	50	42	2.35	0.86
12	250	40	42	2.29	1.88
13	250	40	150	1.85	2.92
14	250	50	60	2.17	3.54
15	250	60	42	1.12	2.44

**^a^** grams of *p*-anisic acid in 100 g of crude extract; ^b^ grams of crude extract from 100 g of dried flowers.

**Table 4 molecules-27-00970-t004:** Analysis of variance of the *p*-anisic acid selectivity from the crude extract.

Source	DF	Seq SS	Adj SS	Adj MS	*F*	*p*
Regression	9	0.000501	0.000501	0.000056	5.79	0.034
Linear	3	0.000252	0.000059	0.00002	2.05	0.226
T	1	0.000244	0.000001	0.000001	0.09	0.773
P	1	0.000004	0.000057	0.000057	5.92	0.059
G	1	0.000004	0.000003	0.000003	0.34	0.587
Square	3	0.000054	0.000054	0.000018	1.87	0.252
T*T	1	0.000007	0.00001	0.00001	1.02	0.359
P*P	1	0.000046	0.000047	0.000047	4.9	0.078
G*G	1	0.000001	0.000001	0.000001	0.1	0.76
Interaction	3	0.000195	0.000195	0.000065	6.75	0.033
T*P	1	0	0	0	0	0.975
T*G	1	0.000072	0.000072	0.000072	7.47	0.041
P*G	1	0.000123	0.000123	0.000123	12.79	0.016
Residual error	5	0.000048	0.000048	0.00001		
Lack-of-fit	3	0.000041	0.000041	0.000014	4.18	0.199
Pure error	2	0.000007	0.000007	0.000003		
Total	14	0.000549				

DF: Degrees of freedom; Seq SS: sequential sum of squares; Adj SS: adjusted sum of squares; *F*: *F*-statistics; *p*: *p*-value. T, P, and G correspond to the variables: temperature, pressure, and medium particle size, respectively. *: the interaction between factors.

**Table 5 molecules-27-00970-t005:** Adjusted and calculated parameters for mathematical models of mass transfer.

**Model**	**Adjusted Parameters**				**Calculated Parameters**	
Crank (R² = 0.9865)	D (m²·s^−^¹)					
	8.424 × 10^−10^					
Sovová (R² = 0.9772)	*Z*	*W*	*x_k_*	*y_r_*	*k_s_* (m·s^−1^)	*k_f_* (m·s^−1^)
	4.721 × 10^−2^	7.753 × 10^−2^	3.532 × 10^−2^	5.368 ×10^−1^	1.206 × 10^−9^	9.684 × 10^−10^
Reverchon (R² = 0.9420)	*t_i_* (s)	*K* (m³·kg^−1^)			*Di* (m²·s^−1^)	*k_TM_* (m·s^−1^)
	1710	5.294 × 10^−3^			1.228 × 10^−12^	5.848 × 10^−4^

**Table 6 molecules-27-00970-t006:** Gradient solvent system used in column chromatography.

Solvent	Ratio (% *v*/*v*)	Subfraction Collected
Hexane	100	1
Hexane:Ethyl acetate	80:20	2
Hexane:Ethyl acetate	60:40	3
Hexane:Ethyl acetate	40:60	4
Hexane:Ethyl acetate	20:80	5
Ethyl acetate	100	6
Ethyl acetate:Dichloromethane	50:50	7
Dichloromethane	100	8
Dichloromethane:Methanol	50:50	9
Methanol	100	10

## Data Availability

Not applicable.

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
