# Peer review of "The Response Surface Optimization of Supercritical CO_2_ Modified with Ethanol Extraction of *p*-Anisic Acid from *Acacia mearnsii* Flowers and Mathematical Modeling of the Mass Transfer"

_molecules, 2022, doi:10.3390/molecules27030970_

Round 1
Reviewer 1 Report
The present paper deals with the extraction of Acacia mearnsii by supercritical carbon dioxide method aiming at receive an extract in good yield with high p-anisic acid content. Antibacterial acivity of the extracts and fractions was measued, and the p-anisic acid content determined by HPLC. Response surface methodology and Box-Behnken design was applied for optimisation of the SC process. It was concluded that the supercritical fluid extraction can be applied for separation an extract containing p-anisic acid in maximum 2.48%. However, the previously uninvestigated A. mearnsii flower was analysed only for one compound (p-anisic acid) and no broader information gained about its chemical composition.
Questions, mistakes?
- The auctor name of the plant and plant family should be given.
- A voucher specimen should be given for plant material used.
- The sentence „The extracts of different Acacia species are used in cosmetics, such as Catechu, Concinna, …..” should be correct „The extracts of different Acacia species, as A. catechu, A. concinna, ….. are used in cosmetics, such as Catechu, Concinna, …..”
- Table 2 containing experimental details should be deleted and its content given in the Materials and methods. HPLC and TLC chromatograms of subfractions (Fig 2, 3) are not of interest, these should be published as Supporting Information.
- Why the antimicrobial activity of the target compound p-anisic acid was not determined? This is a serious deficiency! In the antimicrobial assay, amoxicillin was used as positive control but its MIC value is not given.
- In p-anisic acid „p=para” should be in italic.
- In the Materials and Methods part some description should be significantly reduced, unnecessary details are given in 4.4.1., 4.4.2. Please refer for published method desciption where possible.
- In References the plant names should be in italic; this is a mistake in many times.
Author Response
The answers for the Reviewer 1 (Review report) are in the attached file.

Reviewer 2 Report
The present manuscript (molecules-1532240) is dealing with the extraction of p-anisic acid from A. mearnsii flowers, through supercritical fluid extraction. The manuscript presents important solutions for the environmentally friendly extraction of substances from valuable but little-used raw materials with the characteristics of the mass transfer process. Such a study is extremely useful for reducing the number of tests under the conditions of organizing a production line. The conducted research can be considered reliable, taking into account the statistical processing of data and the evaluation of the composition of extracts with the proposed instrumental and analytical methods. In order to improve the general quality of the work, I suggest only some minor changes:
123 Figure 3 is of noticeably poor quality, it would be advisable to improve it, as well as add the value of the p-anisic acid retention factor Rf and visualize it by TLC.
262 Due to the effect of CO2, the plant material should become more permeable at a load of 278.8 bar and thus increase the solubility and significantly increase the value of the diffusion coefficient. What can explain the extremely low value of the diffusion coefficient (D=8.424x10-10)? In this regard, how to explain the unsatisfactory description of the experimental curve in the area of the extract yield from zero to 2% (Fig.6)?
331 Please indicate the mass in grams of silica gel taken.
350 Did I understand correctly that the acetonitrile content decreases from 80 to 20% during the analysis?
537 The authors used 73 references, of which one is very old (1975), 8 sources are old (before 2000), 28 sources relate to the period 2000 - 2009 and only 36 are younger than 2009, including a reference to the patent.
Author Response
The answers for the Reviewer 2 (Review report) are in the attached file.

Reviewer 3 Report
This article reports the extraction of p-anisic acid (a compound with high industrial interest) from Acacia mearnsii flowers using supercritical fluid extraction. The paper is interesting and innovative in the application of the SFE to this raw material. However, a major revision should be done by authors to improve this manuscript before publication.
Title - Authors should review the title, as the process has been studied using co-solvent and not only using supercritical CO2.
Introduction section - What is the polarity of the target compound? Please, add this information in the introduction section.
Results Section:
Line 106. The authors say that the 7 extracts were analyzed by HPLC, but they do not present the results. Please, add to Table 1 the results obtained in the HPLC analysis
Table 2. What were the criteria for selecting the solvents and the ratio used?
Table 3. Why was only subfraction 4 chosen for antimicrobial analysis? Why didn't the authors study subfraction 5?
Line 135. The authors say that the MIC value found for the extract of subfraction 4 is due to the concentration of p-anisic acid in that subfraction. What is the concentration of p-anisic acid in subfraction 4 and in the other subfractions obtained?
Line 181. If with 120 bar and ethanol as solvent, the obtained global yield was 1.7% and with 240 bar the global yield was 2.49% (both at 40 °C, Table 1). How do the authors explain the fact that they obtained more than 5% yield with a pressure of 200 bar and temperature of 40 °C (Table 4)? Since the authors considered that this small pressure variation does not influence the results obtained. Please see Line 219: This result is also very close to the value found for extraction in the conditions of 300 bar, 40 °C, and 42 mesh (yield of 2.48%) due to the slight variation in the extraction conditions.
Figure 4. Why do authors present the T x P graph if the interaction between these variables had no significant effect on the recovery of the p-anisic acid?
Why did the authors study purification only in the first step of the study? Why did the authors not purify and evaluate the antimicrobial capacity of the extract obtained after the optimization of the extraction?
Line 288. Have the flowers been milled after drying? If yes, please add this information.
Line 297. Was the used raw material milled? What was the particle size of the samples used in the extractions (Step I)? What was the S\F (solvent-to-feed ratio) used in the two steps of the experiments? Please add this information.
Line 303. What is the polarity of p-Anisic acid? Is it a water-soluble compound? What is the polarity of the water under the studied conditions? Justify the choice of this solvent based on its solubility under the conditions studied.
What was the conclusion of the authors about the developed study? The conclusion section is missing.
Author Response
The answers for the Reviewer 3 (Review report) are in the attached file.

Round 2
Reviewer 1 Report
I accept all modifications of the Authors.
Author Response
Reviewer #1 accepted all changes to the manuscript and did not provide additional comments.
Reviewer 3 Report
The authors improved the article and it can be accepted after a minor revisión.
Please consider the suggestions below:
Line 129-130: The sentence “Subfraction 5 was not used in this test due to the similarity between the composition of subfractions 4 and 5 and because subfraction 4 resulted in greater mass” must be rewritten (Poor English).
Lines 309-310: Include the mill specifications, as well as the methodology (equation or model/some reference) used to determine the particle size.
Line 493: It is not Figura but Figure.
Author Response
REVIEWER #3
Comments and Suggestions for Authors
Please consider the suggestions below:
Line 129-130: The sentence “Subfraction 5 was not used in this test due to the similarity between the composition of subfractions 4 and 5 and because subfraction 4 resulted in greater mass” must be rewritten (Poor English).
Answer: The sentence has been rewritten
”We did not use subfraction 5 for the antimicrobial assays due to the lower yield compared to subfraction 4, since both have a similar chromatographic profile.”
Lines 309-310: Include the mill specifications, as well as the methodology (equation or model/some reference), used to determine the particle size.
Answer: We have included specifications of the methodology used in particle size determination. A reference (mathematical model) has also been included.
“Particle size analysis was performed using a set of standard Tyler Series sieves. 100g of the milled plant was added to a set of 6 sieves with 24, 32, 42, 60, 150, and 325 mesh and they were shaken for 15 min using a vibrating stirrer. The fractions with particle sizes of 42, 60, and 150 mesh were selected for the experiments [73].”
- Carvalho, R.N.; Moura, L.S.; Rosa, P.T.V.; Meireles, M.A.A. Supercritical fluid extraction from rosemary (Rosmarinus officinalis): Kinetic data, extract’s global yield, composition, and antioxidant activity. Journal of Supercritical Fluids 2005, 35, 197–204, doi:10.1016/j.supflu.2005.01.009.
Line 493: It is not Figura but Figure.
Answer: We changed "Figura" to Figure in the manuscript text and support information.
